# Impact of the Mining Process on the Near-Seabed Environment of a Polymetallic Nodule Area: A Field Simulation Experiment in a Western Pacific Area

**DOI:** 10.3390/s23198110

**Published:** 2023-09-27

**Authors:** Bowen Li, Yonggang Jia, Zhihan Fan, Kai Li, Xuefa Shi

**Affiliations:** 1College of Oceanography and Space Information, China University of Petroleum, Qingdao 266580, China; 20200111@upc.edu.cn; 2Laboratory for Marine Geology, Qingdao National Laboratory for Marine Science and Technology, Qingdao 266100, China; yonggang@ouc.edu.cn; 3Shandong Provincial Key Laboratory of Marine Environment and Geological Engineering, Ocean University of China, Qingdao 266100, China; fanzhihan@stu.ouc.edu.cn (Z.F.); lk@ouc.edu.cn (K.L.); 4First Institute of Oceanography, Ministry of Natural Resources (MNR), Qingdao 266061, China

**Keywords:** in situ observation, current, sediment resuspension, resuspended sediment transportation, Western Pacific

## Abstract

With the consumption of terrestrial metal resources, deep-sea polymetallic nodule minerals have been widely exploited around the world. Therefore, the environmental impact of deep-sea polymetallic nodule mining cannot be ignored. In this study, for the first time, a field disturbance and observation device, integrated with multiple sensors, is used to simulate the disturbance process of mining on seabed sediments in the polymetallic nodule area of the western Pacific Ocean at a depth of 5700 m. The impact of the process of stroking and lifting on the bottom sediment in the polymetallic nodule area is 30 times higher than that caused by the waves or the current. The time for turbidity to return to normal after the increase is about 30 min, and the influence distance of a disturbance to the bottom bed on turbidity is about 126 m. The time it takes for density to return to normal is about four hours, and the influence is about 1000 m. At the same time, the resuspension of the bottom sediment leads to an increase in density anomaly and salinity. Moreover, suspended sediments rich in metal ions may react with dissolved oxygen in water, resulting in a decrease in the dissolved oxygen content and an increase in ORP. During the observation period, the phenomenon of a deep-sea reciprocating current is found, which may cause the suspended sediment generated by the continuous operation of the mining vehicle to produce suspended sediment clouds in the water near the bottom of the mining area. This could lead to the continuous increase in nutrients in the water near the bottom of the mining area and the continuous reduction in dissolved oxygen, which will have a significant impact on the local ecological environment. Therefore, the way mining vehicles dig and wash in water bodies could have a marked impact on the marine environment. We suggest adopting the technology of suction and ore separation on mining ships, as well as bringing the separated sediment back to the land for comprehensive utilization.

## 1. Introduction

About 10–30% of the seabed is covered with polymetallic nodules, rich in iron, cobalt, nickel, copper, titanium, and other metal elements [1]. This suggests that the polymetallic nodules resource potential in the Clarion-Clipperton Zone (CCZ) of the Pacific Ocean alone is equivalent to 5–10 times the global terrestrial resource reserves [2,3,4]. These metal resources are very precious in the development of modern society. As a result, research on deep-sea polymetallic nodules from the 1960s has been led by developed countries, including the U.S., Germany, and Japan [5,6,7,8]. However, due to economic, technological, political, environmental, and other constraints [9], research on polymetallic nodules has not yet reached the stage of commercial exploitation. Nonetheless, polymetallic nodules are still the most promising for the commercial exploitation of deep-sea metal resources [1,9,10] and are the main representatives of current deep-sea mining research.

Deep-sea polymetallic nodules are generally distributed in abyssal plains with a water depth of 4000–6000 m. The CCZ is most widely distributed in the eastern Pacific, where the average abundance of polymetallic nodules is about 15 kg/m^2^ [11]. It is also distributed in the Indian Ocean, the Cook Islands, and the Peruvian Basin [12,13] with average abundances of 5 kg/m^2^, 5 kg/m^2^, and 10 kg/m^2^, respectively [14]. Polymetallic nodules appear on the surface of the seabed and are mostly spherical, but also take on other shapes, such as those of rods, strips, and cauliflowers [15]. The diameter of the nodules is mostly between 1 and 20 cm, the density is between 1.7 and 3 g/cm^3^, and the burial depth is generally not more than 30 cm, but there are also a small number of polymetallic nodules buried in sediments under 30 cm [16].

The areas in which deep-sea polymetallic nodules occur are relatively deep—generally lower than the carbonate compensation depth (CCD)—and the sediments are mainly composed of siliceous soft clay. The terrain is, on the whole, relatively flat, but there are also small seamounts and hills. These small seamounts and hills affect not only the distribution of nodules [17,18] but also the collection paths and the locations of seabed mining vehicles during nodule mining. The division of early engineering geological units essentially only considered two factors: the seafloor topography and sediment type. The engineering geological conditions of an exploration area were roughly divided over a wide range, but the abundance of nodules, the physical and mechanical properties of the sediments, and the hydrodynamic conditions near the seabed were not considered [19,20,21]. In addition, the spatial distribution of organisms exhibits a certain regularity, which is also related to the distribution of polymetallic nodules [22,23,24,25].

In future commercial mining, the above characteristics will be used as the basis for the division of engineering geological units and mining units in mining areas. A mining vehicle comes into direct contact with surface sediments; the sediments’ physical and mechanical properties not only affect the movement of the mining vehicle, but are also key factors affecting the degree of mining disturbance. Because deep-sea polymetallic nodule mining is a digging process, the environmental impact of polymetallic nodule mining mainly comes from the process of digging and tailings discharge [26], and the impact of bottom plume generated during the digging process accounts for more than 95% of the total impact. Mining operations disturb seabed sediments, change the living environment of benthic organisms, increase the concentration of suspended sediments in water, and affect the chemical properties of water bodies [4,27]. Some researchers even believe that deep-sea mining will have an impact on global climate change [28], but most researchers still focus on the impact of deep-sea mining on organisms [27,29].

Research on seabed mining vehicles mainly focuses on the questions of how to normally walk underwater, how to collect nodules more effectively, and how to realize the positioning of mining vehicles in the presence of interference. In fact, the problem of mining vehicles traveling in nodule areas is highly complex and involves coupled processes between seabed mining vehicles and sediments [30,31]. Aside from the sealing and compression issues caused by water depth [27], from a sedimentary perspective, the shallow sediments in areas of deep-sea polymetallic nodule occurrence are very weak. On the one hand, they cannot provide a high bearing capacity, and, on the other hand, they cannot provide sufficient shear resistance for the forward movement of minecarts [32]. In addition, the presence of nodules on the surface of the seabed adds many uncertainties to the properties of sediments [33], and rapidly deposited sediments will quickly cover the original seabed, change its morphological structure, and affect the activities of benthic organisms [33,34]. Therefore, designers need to consider a wider range of issues. 

Based on the results of the current simulated mining experiment in the polymetallic nodule area, 5–15 cm of sediment on the seabed surface could be removed during mining [35]. It is estimated that in the commercial mining process, 2.5–5.5 t of sediment will be disturbed and resuspended for every 1 t nodule mined [36]. Disturbance experiments conducted by mining vehicles show that both the sediment compacts at the rut and the sediment redeposited outside the rut could have great changes in physical and mechanical properties, and the pore structure of the sediment inside and outside the disturbed area is still different from that in the undisturbed area after 26 years [37]. With the development of technology, in order to better assess the impact of deep-sea mining, submarine camera [38] harmonic reflection inversion [39] was developed to obtain the harmonic reflection intensity of submarine video images.

In addition to the latest mining-vehicle-based method of seabed mining, other seabed mining methods have been devised in various countries, such as the submarine–drag bucket mining system, the continuous line bucket mining system, the shuttle vessel mining system, and so on [40]. However, these mining methods all involve the extraction of the bottom bed and affect the subsea environment nearby [41]. At present, there are no observational reports on the environmental impact of simulated mining processes in the Western Pacific polymetallic nodule mining area. In this paper, a device with sensors is used to stroke, draw, and imitate the digging action of a mining vehicle. The changes and effects of the sediments suspended in the water body at a depth of 5700 m during the process were analyzed in order to provide more detailed data for the realization of deep-sea mining.

## 2. Methods

### 2.1. Experiment Site

This study used a device to simulate the process of the stroke and lifting of seabed mining vehicles. Equipment was mounted on the device to observe sediment resuspension and changes in hydrodynamics during the process of the stroke and lifting of the device in a polymetallic nodule area at a depth of 5700 m (153.41° E, 7.48° N) in the Western Pacific Ocean from 8 June 2021 at 21:00 to 17:00 on 9 June (Figure 1). The terrain of the experimental site is flat, and the surrounding depth does not significantly change, which is conducive to the mining of polymetallic nodules (Figure 1: the trace in the middle of the terrain scan map is the midline of the ship). The device was located at the point GC1. The diameter of the nodules ranges from 3 to 10 cm, with an average diameter of about 5 cm. The sediments above the base of the study area are mainly deep-sea clay and calcareous ooze [42].

### 2.2. In Situ Observation Device

The device and its carrying equipment are shown in Figure 2. The total height of the device was 2.7 m, the total weight in the air was 900 kg, and the total weight in the water was 350 kg. Deployment was used to simulate the impact of mining vehicles stroking in the deep sea. Because the pressure applied by the device to the bottom bed is about 3610 Pa, its impact is close to that of the “Kunlong” mining vehicles on the bottom bed [43,44]. The distance between the two legs of the device is similar to the distance between the two tracks of the “Kunlong” mining vehicle, at about two meters [44].

The device integrates a variety of sensors, such as temperature, salt, pressure, dissolved oxygen, ORP, and turbidity sensors, a submarine camera, and a 2M HZ ADCP (acoustic Doppler current profiler). The accuracy of the temperature and salt sensors was 0.002 degrees; the salinity accuracy was 0.003 ms/cm. The accuracy of the pressure sensor was 0.05% of the water depth. The accuracy of the optical dissolved oxygen sensor was 5%. The accuracy of the ORP sensor was 0.01 V. The accuracy of the turbidity sensor was 2%. The memory of the submarine camera could be continuously stored for 20 h. The acquisition frequency of the ADCP was 1 s/time, the blind area was 0.5 m, and the distance was 0.1 m per layer.

### 2.3. Experimental Process

The experiment used the above device to simulate the impact of mining vehicles landing and placing on the seabed for a certain time and lifting. The device was placed using a cable and entered the water at 16:09 on 8 August, hitting the bottom at 21:27 with a dropped velocity of 1 m/s. Two minutes later, the ship wobbled, causing the device to wobble. The device was linked to the ship by a cable during the observation period, as the mining vehicle was linked to the mining vessel by a pipeline [45]. The lifting (lifting the bottom) of the device began at 18:39 on 9 August, and the device emerged from the water at 20:24.

## 3. Results

### 3.1. The Change Process of the Current’s Direction near the Bottom Bed

According to Figure 3, the current direction of the near-bottom bed changed between 8:00 and 12:00 on 9 August; the average degree of current direction at different water depths before and after the change is shown in Figure 4. They are all different, but the current direction before 08:00 is similar to the current direction 10 cm above it after 12:00. It may be that, in the four hours during which the current changed, from 8:00 to 12:00 on 9 August, the upward propagation of the water mass was 10 cm above 0.2 m from the seabed, but the current direction at the height of 0.2 m is similar to that at the height of 0.1 m. This could be due to the terrain causing the destruction of water masses. The interaction of these water masses with the terrain can cause the flat seabed to become resuspended [46,47,48]. The average current velocity in the horizontal direction is 7 cm/s in its normal sea conditions (from 11:00 on 8 August to 8:00 on 9 August) (Figure 3).

### 3.2. Process of Variation in Suspended Sediments near the Bottom Bed

In total, 21 h, from 21:00 on 8 August to 18:00 on 9 August 2021, were analyzed. Obvious sediment resuspension was detected during the device’s stroking and lifting, and the turbidity in the water body also increased by 80 NTU and 40 NTU, respectively (Figure 5). In addition, during the observation period, the bottom sediments were only resuspended during the rapid increase in the current velocity during the change in the current’s direction from 8:00 to 12:00 on 9 August (Figure 5).

The changes in the suspended sediments could also be clearly observed with the camera (Figure 6). When the observation device began stroking on 8 August, 9:27, sediments were obviously resuspended (Figure 5 and Figure 6a,b). After 70 min, the water body gradually became clear (Figure 6c), and, after about two hours, the water body’s turbidity basically returned to the levels detected before the observation device’s stroke (Figure 6d). The background value of turbidity was about 0.2 NTU (Figure 5). On 9 August, at 8:48, the sediment resuspension caused the water body to become cloudy again, and the process of the sediment resuspension lasted about 45 min (Figure 6e–g). However, the turbidimeter did not observe a significant change in turbidity, possibly because the height of the resuspension was lower than the observation position of the turbidimeter (0.6 m from the bottom surface).

The amount of sediment resuspended during the stroking and lifting of the device accounted for 47% of the total amount of resuspension. The background value of the apparent turbidity was 0.2 NTU, and it was not considered resuspension if the turbidity was less than 0.2 NTU, so the amount of sediment resuspension during stroking and lifting accounted for 99% of the total resuspension amount.

## 4. Discussion

### 4.1. Effects of Sediment Resuspension on Sediment Flux

In this study, the turbidity, measured with a turbidity meter, at 60 cm from the bottom bed and the current velocity at different distances from the bottom bed, measured with an ADCP, were used to calculate the sediment flux values at different heights from the bottom bed (Figure 7 and Figure 8). The trends in the variations in the sediment fluxes at different heights during the observation period were found to be consistent (Figure 7 and Figure 8). The total sediment flux before the change in the current direction was about 1.6 times that after the change, and the average sediment flux per unit of time was about 1.2 times that after the change. Considering that the suspended sediment flux data when stroking were more complete and larger than the suspended sediment flux data when leaving the ground, the average values of sediment flux per unit of time before and after the change in the current direction were close to one another.

The sediment flux increased during the stroking and lifting of the device, with a maximum value of 1000 NTU × cm/s. In addition, the suspended sediment flux only significantly increased around 6:00 and 11:00 on 9 August (Figure 7 and Figure 8). This is similar to the sediment resuspension process in response to turbidity values. The suspended sediment flux during the stroking and lifting of the device accounted for 27% of the total suspended sediment flux. If the background value of the apparent turbidity is 0.2 NTU, the suspension when the turbidity is less than 0.2 NTU is not considered. The suspended sediment flux accounted for 97% of the total suspended sediment flux. Thus, the impact of the mining vehicle’s digging (stroking and lifting) on the bottom sediments of the polymetallic nodule area was 30 times larger than that of the waves or current. The impact of the mining vehicles is the main control factor for sediment resuspension and the migration process.

### 4.2. The Effect of Sediment Resuspension on Salinity and Water Concentration Anomaly

After the device landed and sediment resuspension occurred in the bottom bed, the value of density anomaly in the water body rapidly rose and then decreased; then, it continued to rise with the increase in the salinity value (Figure 5). The rapid increase and then decrease in density anomalies after the resuspension of the bottom sediment may have been caused by the resuspension and deposition of sediments. Then, the density anomalies continuously increased with the increase in salinity, perhaps because minerals in the resuspended sediment were released into the water body. In order to verify this conjecture, first, the relationship between the density anomalies and salinity was calculated, and the two were found to be positively correlated. The correlation coefficient was 0.78 (Figure 9). Except for the rapid increase in density anomalies at the beginning of the observation period, the two were basically linearly correlated. Thus, the later changes in density anomalies were mainly caused by salinity.

On this basis, the correlation between density anomalies and salinity was used, the influence of salinity on density anomalies was excluded, and the change curve of density anomalies and turbidity was plotted (Figure 10). An increase in density anomalies was found to occur after resuspension, but there was a rapid decrease after the deposition of the suspended sediments. A rapid rise and fall in density anomalies was caused by the resuspension of bottom sediments. Moreover, the increases in salinity were also mainly caused by the resuspension of sediments due to the stroke and excavation of the device. The time it took for turbidity to return to normal after the rise was about 30 min, while the time it took for density to return to normal was about four hours. Therefore, the digging (stroking and lifting) of mining vehicles may cause the precipitation of metal particles and their re-entry into the water body, thus causing environmental damage to the areas in which polymetallic nodules are deposited. Based on the average current velocity of about 7 cm/s, it can be estimated that the influence distance of a disturbance to the bottom bed on turbidity is about 126 m, and the influence distance to the density is about 1000 m. In the process of conducting in situ observations, the ORP (oxidation/reduction potential) continues to increase while the dissolved oxygen continues to decrease (Figure 5). This may also be related to the reaction of resuspended metal-rich ions with dissolved oxygen in the water body.

### 4.3. Effects of the Deep Reciprocating Current on the Near-Seabed Environment

During the observation period, the direction of the deep-sea reciprocating current near the seabed, with a distance of 0.6 m from the seabed, changed from northwest to southeast. The transport time to the northwest is almost the same as the transport time to the southeast. The east–west current velocity is relatively balanced, and the sum of the east–west current velocity during the observation period is about −18 m (the water particle moves to the west by 18 m). The southbound current velocity was larger than the northbound current velocity, and the southbound water particle movement was about twice that of the northbound water particle movement during the observation period; the southbound net movement of the water particles was about 800 m. From the time of impact to around 14:00 on 9 August, the net north–south movement of the particle point was 0, and the net east–west movement was about 200 m (Figure 11). Therefore, the increase in turbidity from 8:00 to 12:00 on 9 August was not caused by the transport of sediment resuspension caused by the stroke of the device but may have been caused by the increase in the current velocity during the current direction conversion (Figure 3). As the above analysis shows that, during the observation period, the net movement of water quality points in the polymetallic nodule deposition area may be 0 in the north–south and east–west directions, the sediments continuously suspended during the process of mining polymetallic nodules may move back and forth in the mining area, and even combine with each other to form suspended sediment clouds.

## 5. Conclusions

By observing the impacts of the process of the stroke and lifting of mining vehicles on the near-bottom-bed environment of a polymetallic nodule deposition area in the Western Pacific Ocean, we detected the phenomenon of a tide-like change in the current direction (deep-sea reciprocating current) near the seabed in the Western Pacific Ocean. However, this phenomenon did not involve slack water during the transition of the ebb and flow tides. The tide-like change in the current direction was possibly caused by the downward propagation of the water mass.

By simulating the process of the stroke and lifting of mining vehicles with a device, we found that the impact of the process of deployment and lifting on the bottom sediment in the polymetallic nodule area was 30 times greater than that caused by the decrease in the current velocity after the change in the current direction; these were the main control factors for sediment resuspension. Sediment resuspension induced a rapid increase and then a decline in the density anomaly. After that, the density anomaly continuously increased with the increase in the salinity value, which was the result of the continuous entry of the minerals in the suspended sediments into the water body. The time it took for turbidity to return to normal after the rise was about 30 min, and the influence distance of a disturbance to the bottom bed on turbidity was about 126 m. The time it took for density to return to normal was about four hours, and the influence was about 1000 m.

Although the impact strength of this time may be greater than the formation sediment disturbance caused by polymetallic nodule mining, the existence of deep-sea reciprocating currents may lead sediments that are resuspended in the process of polymetallic nodule mining to move back and forth in the mining area, and even to combine with each other to form suspended sediment clouds. As a result, the water density and salinity in the mining area continue to rise and the dissolved oxygen concentration decreases, which causes significant harm to the ecology of the near-bottom bed in the polymetallic nodule deposition area. Based on the above reasoning, the way in which mining vehicles dig and wash in water bodies could have a significant impact on the marine environment. We recommend adopting the technology of suction and ore separation on mining ships, and bringing the separated sediment back to the land for comprehensive utilization.

In this study, a device weighing 350 kg in water was used to simulate the process of the stroke and lifting of a mining vehicle. The weight of the device and the contact area with the bottom bed were different from those of a real mining vehicle, so only qualitative results could be obtained in this simulation experiment. In order to obtain accurate results from the assessment of the impact of the process of the stroke and lifting of mining vehicles on the near-bottom-bed environment, more experiments that evaluate or monitor the trial mining processes of mining vehicle prototypes are needed.

## Figures and Tables

**Figure 1 sensors-23-08110-f001:**
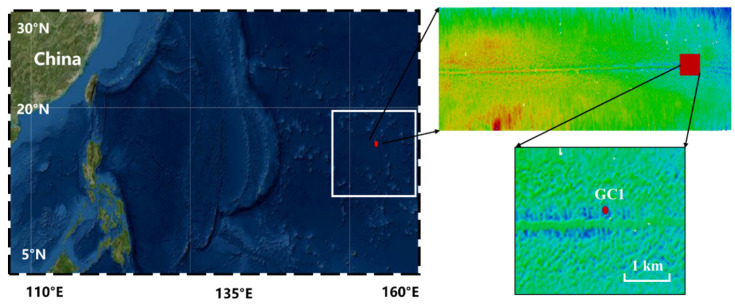
The in situ observation site was located in a polymetallic nodule area (153.41° E, 7.48° N) in the Western Pacific Ocean.

**Figure 2 sensors-23-08110-f002:**
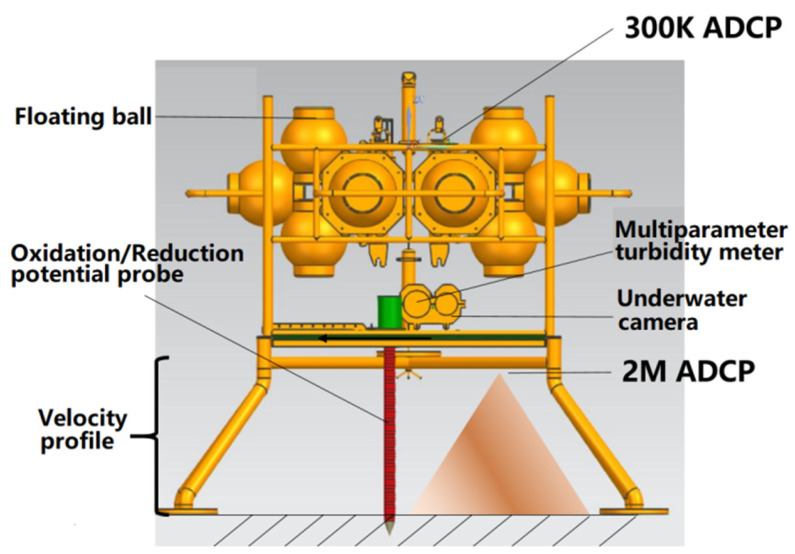
The device and the carrying equipment were used to collect environmental data.

**Figure 3 sensors-23-08110-f003:**
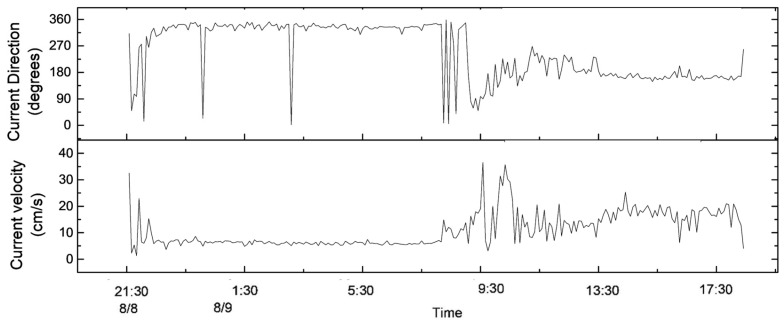
The changes in the direction and velocity of the current, tested using an ADCP near the bottom seabed (the distance from the seabed is 0.6 m).

**Figure 4 sensors-23-08110-f004:**
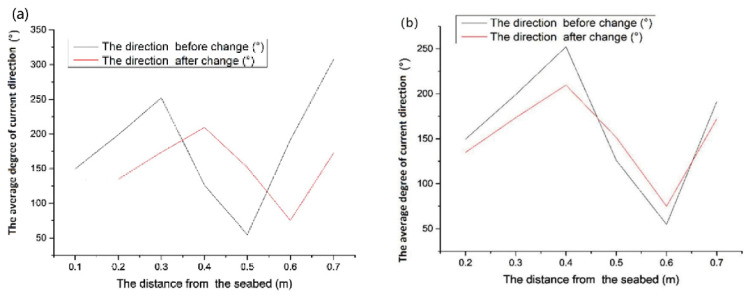
The change in the direction of the current near the bottom bed from 8:00 to 12:00 on 9 August; (**a**) the average degree of the current’s direction before and after the change, (**b**) a comparison of the average degree of the current’s direction before and after the change.

**Figure 5 sensors-23-08110-f005:**
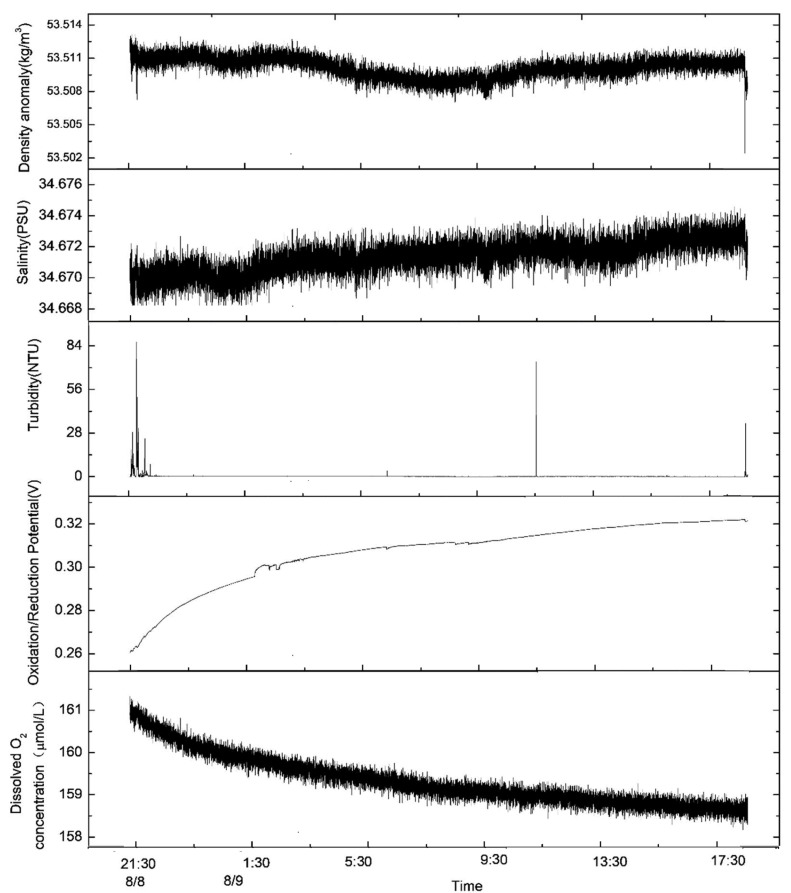
Data from in situ observations from 21:00 on 8 August to 18:00 on 9 August 2021.

**Figure 6 sensors-23-08110-f006:**
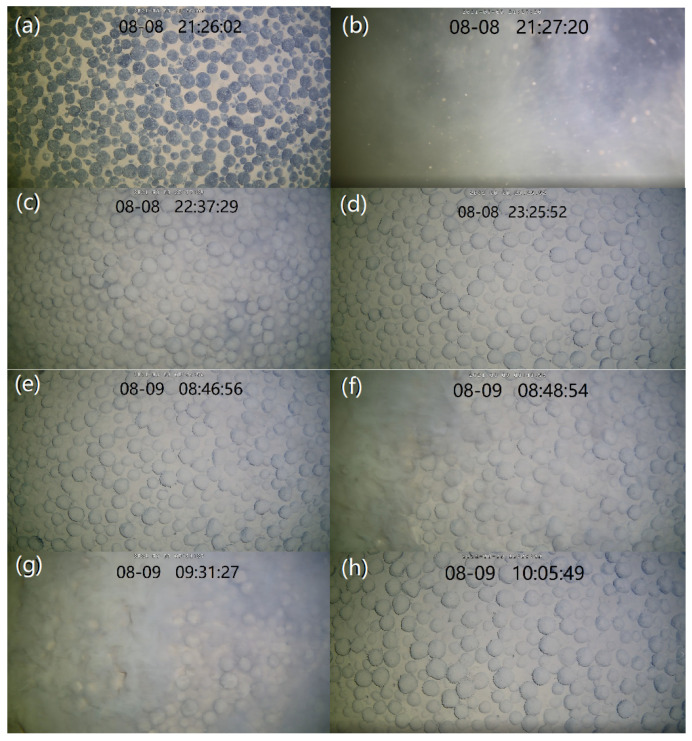
The sediment resuspension induced by the stroke of the device and the current was recorded with a camera: (**a**) before the stroke of the device; (**b**) the resuspension induced by the stroke; (**c**) 70 min after the stroke; (**d**) 120 min after the stroke; (**e**) before the increase in the current velocity; (**f**) the resuspension induced by the current; (**g**) 40 min after the increase; (**h**) 80 min after the increase.

**Figure 7 sensors-23-08110-f007:**
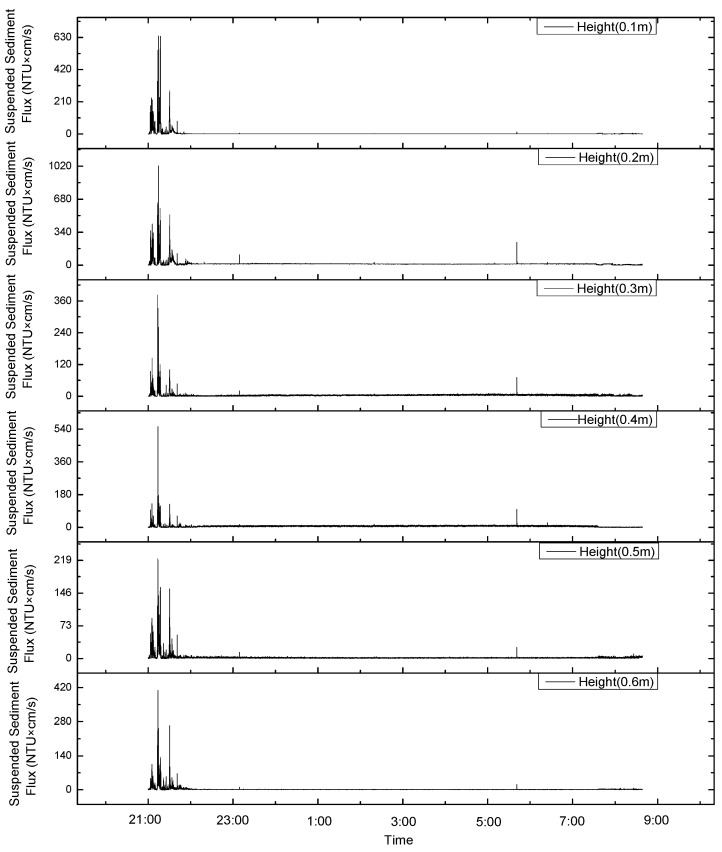
The trends in variations in the sediment fluxes at different heights during the observation period were consistent before the change in the current’s direction.

**Figure 8 sensors-23-08110-f008:**
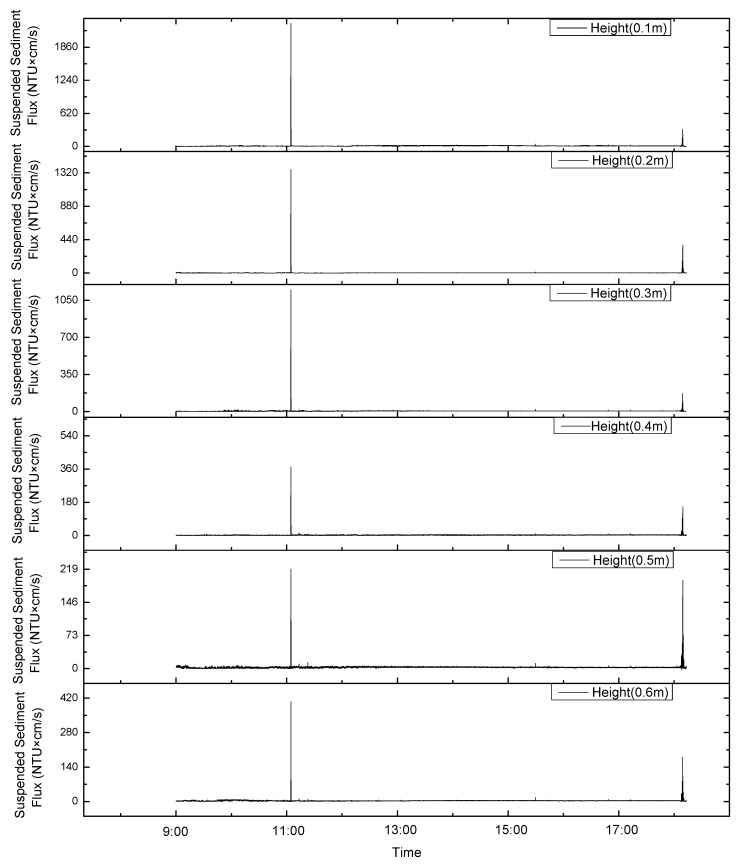
The trends in variations in sediment fluxes at different heights during the observation period were consistent after the change in the current’s direction on 9 August.

**Figure 9 sensors-23-08110-f009:**
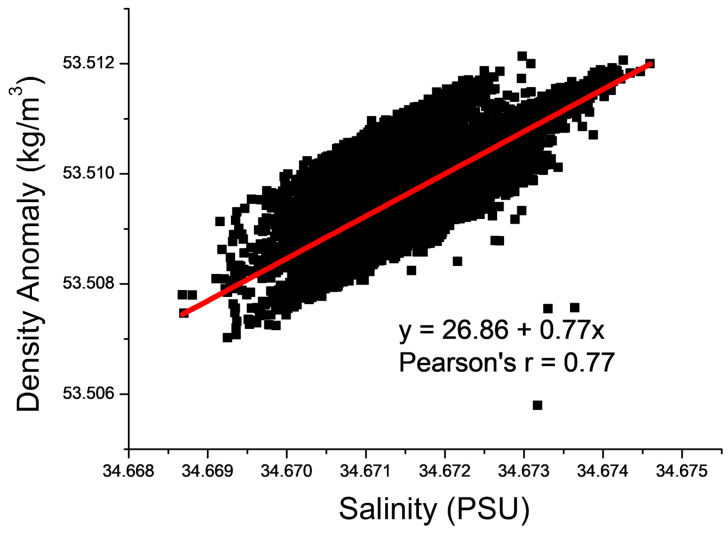
The density anomaly and salinity were positively correlated, and the correlation coefficient was 0.78.

**Figure 10 sensors-23-08110-f010:**
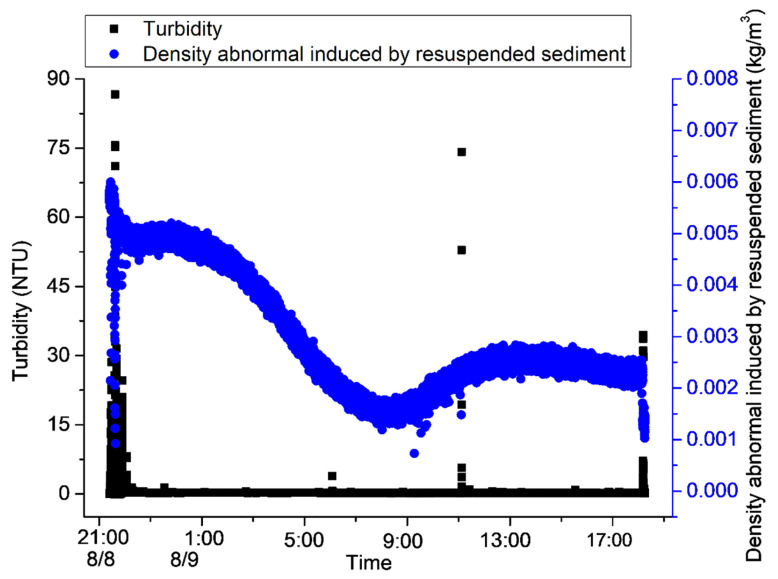
The increase in density anomaly after resuspension.

**Figure 11 sensors-23-08110-f011:**
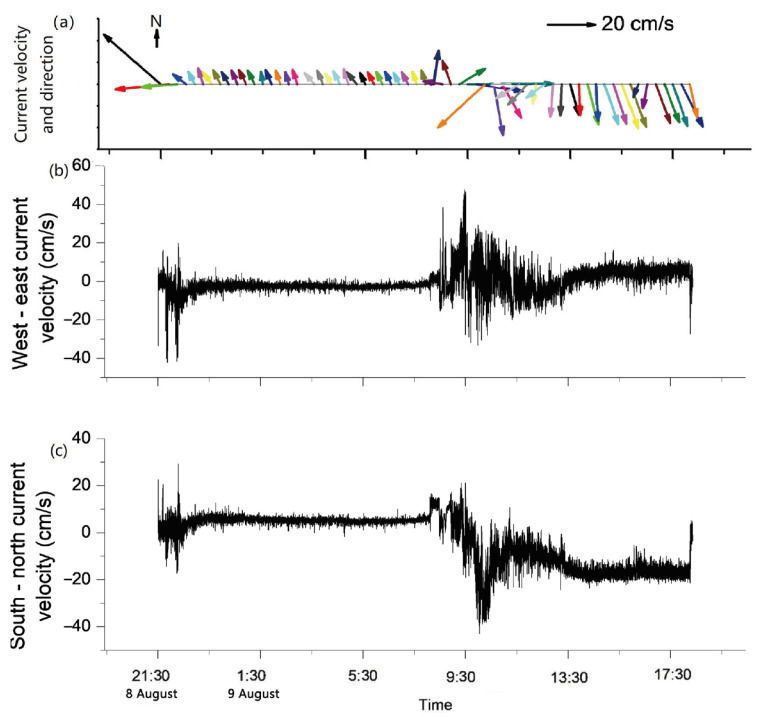
The current velocity and direction (**a**) are decomposed into the west–east direction (west is negative, east is positive) (**b**) and south–north direction (south is negative, north is positive); (**c**) current velocity near the bottom seabed (the distance from the seabed is 0.6 m).

## Data Availability

The data presented in this study are available on request from the corresponding author. The data are not publicly available due to the data takes up too much space.

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
