# Peer review of "Impact of the Mining Process on the Near-Seabed Environment of a Polymetallic Nodule Area: A Field Simulation Experiment in a Western Pacific Area"

_sensors, 2023, doi:10.3390/s23198110_

Round 1
Reviewer 1 Report
Line 4: ... : A field Simulation Experiment in Western Pacific Mining Area
Line 17-18 and in the overall statement: The mechanism of the disturbance on deep-seabed sediment was not described clearly: the way of disturbance, "stroke" and "drawing", is hard to understand. Fig. 2 does not show how the sediment was disturbed. Probably, the device was towed by cable, then a proper word could be "towing", not drawing. About the "stroke", the authors have to show what is the stroke and how the stroke was performed.
Line 168: The average current velocity of 7 cm/s is not shown in Fig. 5, and in Fig. 3 it does not exceed over 2-3cm/s. Authors should show the measured result.
Line 173-175: What is the difference of (a) and (b) of Fig. 4? One, (a) or (b) will be enough.
Line 206-208: The meaning of the sentence is hard to catch. Please, correct the sentence to be easy for understanding.
Check and correct the errors in the references.
Line 225-229: A mining process will require "sweeping" of nodules. Landing and lifting of mining machine are only limited procedures in mining. They can not be the main control factors.
The authors showed importance measurement data on the recovering times of turbidity and density anomaly in Line 254-258. In spite of the meaningfulness, those will be decisively affected by the scope and the way of the disturbance. As the authors themselves cited in Line 80-82. For an instance, if a hydraulic suction is adopted for excavation, the precipitation and re-entry of metal particles in the surrounding water could be dramatically reduced. So, landing and lifting can not be the main control factors in mining operation. Please, correct the statement.
There are lots of errors in grammar and improper words and expressions. A strong correction is needed.
Some of them are as following:
anomalies => anomaly
at different depths => at different heights
Line 103: mining vehicle seabed mining => seabed mining vehicle
Line 235: number of density anomalies => value of density anomaly
Line 252: grammar error
In Fig. 10: Density abnormal => Density anomaly
Author Response
Point 1: Line 4: ... : A field Simulation Experiment in Western Pacific Mining Area.
Response 1: Thank you for your suggestion. I have revised it in Line “4”.
Point 2: Line 17-18 and in the overall statement: The mechanism of the disturbance on deep-seabed sediment was not described clearly: the way of disturbance, "stroke" and "drawing", is hard to understand. Fig. 2 does not show how the sediment was disturbed. Probably, the device was towed by cable, then a proper word could be "towing", not drawing. About the "stroke", the authors have to show what is the stroke and how the stroke was performed.
Response 2: Thank you for your suggestion. The mining of seabed polymetallic nodules is essentially a digging action. A digging action can be divided into two processes: stroking the bottom bed and drawing. In this experiment, the impact of the experimental device on the bottom is used to simulate stroking the bottom bed, and the process of the experimental device leaving the ground is used to simulate the drawing process. The description of these specific processes is presented in the Lines “ 156-166”.
Point 3: Line 168: The average current velocity of 7 cm/s is not shown in Fig. 5, and in Fig. 3 it does not exceed over 2-3cm/s. Authors should show the measured result.
Response 3: Thank you for your suggestion. I have revised the description of the current vellocity in Lines “180-181” and show the observed data in Fig.3. The average current vellocity here is basically the average current velocity in its normal sea conditions (from 11:00 on Aug. 8 to 8:00 on Aug. 9).
Point 4: Line 173-175: What is the difference of (a) and (b) of Fig. 4? One, (a) or (b) will be enough.
Response 4: Thank you for your suggestion. Figure (a) shows real observational data. Figure (b) changes the current direction from 0.1 m to 0.2 m, from 0.2 m to 0.3 m, and so on, in order to show that the change in current direction may be caused by water mass propagation. This makes it easier to compare the current direction before and after the change. If you think that this change can be seen in figure (a), and the comparison in Figure (b) is unnecessary, we can also delete figure (b).
Point 5: Line 206-208: The meaning of the sentence is hard to catch. Please, correct the sentence to be easy for understanding.
Response 5: Thank you for your suggestion. I have revised the sentence in Lines “ 216-219”. The revised sentences is as follows:
In this study, the turbidity, measured with a turbidity meter, at 60 cm from the bottom bed and the current velocity at different distances from the bottom bed, measured with an ADCP, were used to calculate the sediment flux values at different depths from the bottom bed (Fig. 7, 8).
Point 6: Check and correct the errors in the references.
Response 6: Thank you for your suggestion. I have revised the references.
Point 7: Line 225-229: A mining process will require "sweeping" of nodules. Landing and lifting of mining machine are only limited procedures in mining. They can not be the main control factors.
Response 7: Thank you for your suggestion. The landing and lifting are mainly meant to simulate the action of digging. The main purpose of this sentence is to show that the impact of the mining vehicles is greater than the impact of the hydrodynamic. I have revised the sentence in Lines “240-244 ”. The revised sentences is as follows:
Thus, the impact of the mining vehicle's digging (stroking and drawing) on the bottom sediments of the polymetallic nodule area is 30 times larger than that of the waves or current. The impact of the mining vehicles is the main control factor for sediment resuspension and the migration process.
Point 8: The authors showed importance measurement data on the recovering times of turbidity and density anomaly in Line 254-258. In spite of the meaningfulness, those will be decisively affected by the scope and the way of the disturbance. As the authors themselves cited in Line 80-82. For an instance, if a hydraulic suction is adopted for excavation, the precipitation and re-entry of metal particles in the surrounding water could be dramatically reduced. So, landing and lifting can not be the main control factors in mining operation. Please, correct the statement.
Response 8: Thank you for your suggestion. I have revised the sentence in Lines “272-275”. The revised sentences is as follows:
Therefore, the digging (stroking and drawing) of mining vehicles may cause the precipitation of metal particles and their re-entry into the water body, thus causing environmental damage to the areas in which polymetallic nodules are deposited.
Point 9: Comments on the Quality of English Language
There are lots of errors in grammar and improper words and expressions. A strong correction is needed.Some of them are as following:
anomalies => anomaly
at different depths => at different heights
Line 103: mining vehicle seabed mining => seabed mining vehicle
Line 235: number of density anomalies => value of density anomaly
Line 252: grammar error
In Fig. 10: Density abnormal => Density anomaly
Response 9: Thank you for your suggestion. I have revised these sentences. And the English language has been revised. The related certificate is in the attachment.

Reviewer 2 Report
The proposed paper may be published since it contains an important contribution to the studies on a potenential

Author Response
The proposed paper may be published since it contains an important contribution to the studies on a potential impact of deep sea mining on environment. Probably some language improvements are needed and editorial corrections. The authors should however consider to improve before publication two following, very important issues:
Point 1: The scope of the paper should be in general more focused on the design of the experimental process and the discussion of the relevance of the used device and its equipment for measuring the disturbance on the seabed. The profile of the journal “Sensors” deals more with the hardware and software of the sensoring process itself than the subject of the process. In its present form the paper is profiled rather more on the results and thus it is probably more suited for the publication in one of the journals related to the environmental aspects. This is very important and should be considered by the editors as well.
Response 1: Thank you for your comments. I submitted the paper to the special issus “Observation of Marine Sedimentology” in the journal “Sensors”. The special issue welcome contributions in all areas of marine sedimentological observation and based on various sensors, such as acoustics, visible spectrum, laser, radar, SAR, electricity, thermodynamics, sonar, seism, and so on. These include, but are not limited to, the following: remote sensing, observation of sedimentary dynamic environment, observation of suspended particle concentration; identification of submarine sediment types, exploration of seafloor sedimentary strata; data interpretation model. So I submitted the paper about the observation of sedimentary dynamic environment and observation of suspended particle concentration in in Western Pacific to the special issue. The related information in the website https://www.mdpi.com/journal/sensors/special_issues/9C31F89JJ2.
Point 2: In its present form the authors should consider including results and information on the disturbance tests that were carried out in the past for example the Japanese Jet Experiment or Interoceanmetal works on BIE in the Clarion Clipperton Zone. There is a lot of information available on these tests in scientific papers and journals. But if authors include this information the paper will be even more focused on environmental aspects rather than the profile that is intended to be covered by the papers published in Sensors. I am not sure but probably the paper should be rather considered for publication in journals related to the environmental aspects rather
Response 2: Thank you for your suggestion. I have added the results of other simulated mining experiments, as well as the newly added observation instruments for this work in Lines “100-110”. And the English language has been revised. The related certificate is in the attachment.

Round 2
Reviewer 1 Report
The authors' understanding about the mining process of PMN as a "digging" action is NOT correct. Harvesting or collecting with moving action on the seabed will be the proper understanding. Therefore, the experimental device, which was landed and placed on the seabed for a certain time and recovered, can not represent the mining process.
That is the reason why a change of the title was recommended.
No comments
Author Response
Thank you for your comments.
I have revised "The mining of seabed polymetallic nodules is essentially a digging action. A digging action can be divided into two processes: stroking the bottom bed and drawing. In this experiment, the impact of the experimental device on the bottom is used to simulate stroking the bottom bed, and the process of the experimental device leaving the ground is used to simulate the drawing process." to "The experiment uses the above device to simulate the impact of mining vehicle landing and placing on the seabed for a certain time and lifting" in the “2.3. Experimental process” in Lines "152-153".
And the "drawing" in the paper has been changed to "lifting".
Reviewer 2 Report
Thank you for submitting the improved manuscript and introducing the changes according to my remarks. I consider that the paper can be published if the special issue editors accept the relevance of the content for the sope of the Sensor journal.
Author Response
Thank you very much for your comments on improving the quality of this paper.